# Shedding Light on the Direct and Indirect Impact of the COVID-19 Pandemic on the Lebanese Radiographers or Radiologic Technologists: A Crisis within Crises

**DOI:** 10.3390/healthcare9030362

**Published:** 2021-03-23

**Authors:** Rasha Itani, Mohammed Alnafea, Maya Tannoury, Souheil Hallit, Achraf Al Faraj

**Affiliations:** 1Department of Radiologic Sciences, Faculty of Health Sciences, American University of Science and Technology (AUST), Beirut 1100, Lebanon; Rasha_Itani@outlook.com (R.I.); mtannoury@aust.edu.lb (M.T.); 2Department of Radiological Sciences, College of Applied Medical Sciences, King Saud University, Riyadh 11433, Saudi Arabia; alnafea@ksu.edu.sa; 3Faculty of Medicine and Medical Sciences, Holy Spirit University of Kaslik (USEK), Jounieh 1200, Lebanon; souheilhallit@hotmail.com; 4INSPECT-LB: National Institute of Public Health, Clinical Epidemiology and Toxicology, Beirut 1100, Lebanon

**Keywords:** COVID-19, radiographers, radiology and medical imaging, safety protocols, social and economic consequences

## Abstract

With the novel coronavirus disease 2019 (COVID-19) pandemic, the need for radiologic procedures is increasing for the effective diagnosis and follow-up of pulmonary diseases. There is an immense load on the radiographers’ shoulders to cope with all the challenges associated with the pandemic. However, amidst this crisis, Lebanese radiographers are also suffering from a socioeconomic crisis and record hyperinflation that have posed additional challenges. A cross-sectional study was conducted among registered Lebanese radiographers to assess the general, workplace conditions, health and safety, mental/psychologic, financial, and skill/knowledge development impacts. Despite applying an adapted safety protocol, institutions are neither providing free RT-PCR testing to their staff nor showing adequate support for infected staff members, thus causing distress about contracting the virus from the workplace. Aggravated by the deteriorating economic situation that affected the radiographers financially, they additionally suffer from severe occupational physical and mental burnout. Regardless of that, they used their free time during the lockdown for skill/knowledge development and have performed many recreational activities. This cross-sectional study highlighted the different ways the pandemic has impacted the radiographers: physically, psychologically, and financially. It aimed to shed light on what these frontline heroes are passing through in the midst of all these unprecedented crises.

## 1. Introduction

The outbreak of the SARS-CoV-2 virus in December 2019 through the increasing number of patients suffering from a new form of “viral pneumonia” and the declaration of the coronavirus disease 2019 (COVID-19) pandemic by the World Health Organization (WHO) have flipped the entire world upside down [1]. Despite having less fatality rate (3.4%) and milder symptoms than its precursors, SARS-CoV (9.6%) and MERS-CoV (35%), the new SARS-CoV-2 has faster human-to-human transmission [2], which has been clearly shown by its ability to infect over 83 million people around the globe within its first year [3]. By the end of the year 2020, several potential vaccines (i.e., Pfizer/BioNTech, Moderna, Oxford-AstraZeneca, Sputnik V, etc.) have cleared phase III trials and were approved by health authorities for emergency use [4]. Many countries around the globe have already started their vaccination campaigns, with high hopes of slowing down the spread of the disease. Despite showing more than 90% efficiency in clinical trials, there has been no concrete evidence thus far concerning possible adverse effects and long term immunity, as well as the mechanism of action in the elderly and those suffering from underlying health conditions [5]. Additionally, the sudden emergence of numerous strains worldwide, such as the B.1.1.7 (United Kingdom) and B.1.1.28 (South Africa) linages, added to their divergent mutations, the little known information about their severity, transmission, and resistance have all put the fate of the newly proposed treatments and vaccines at stake [6,7].

In addition to the frightening infection and deaths worldwide, COVID-19 has taken its toll on the global economy. With increasing infections and multiple forced lockdowns, many businesses, manufacturing companies, and organizations reduced their activities, sales, and overall productions, which, in turn, slowed down the global economy until it almost came into a “freeze” [8]. The global economic knockout has drastically affected healthcare workers, particularly the radiology department. With the increasing need of intensive care unit (ICU) beds, medications, and personal protective equipment (PPE) on one hand, and the reduction in admissions on the other hand due to fear of contracting the virus, hospitals are struggling to maintain their revenues [9]. Radiology departments worldwide are experiencing a significant drop in imaging volumes, especially screening services for breast and lung cancer, after the American College of Radiology (ACR) and Centers for Disease Control and Prevention (CDC) implemented several guidelines, some of which included postponing and rescheduling non-urgent patient visits [10].

As for Lebanon, a 10,452 Km^2^ country located in the Middle East, the battle with controlling the viral spread is strenuous. With the first COVID-19 infection reported on 21 February 2020, Lebanon has faced many obstacles that hindered the ability to slow the spread of the virus among its inhabitants [11]. Despite the fact that Lebanon was not the pioneer in healthcare according to the WHO’s report in 2000, holding the rank 91 [12], the country underwent various changes, reforms, and advancements to be ranked as the 23rd country worldwide in 2018 according to Bloomberg’s Healthcare Efficiency Index [13]. However, according to the same index, the country has later on declined and ranked 48th in 2020 amid the pandemic, since Lebanon was less prepared for such a pandemic compared to other countries. Unfortunately, this efficient healthcare system is not free for its citizens. This has led many Lebanese people suffering from COVID-19 symptoms to skip performing necessary diagnostic tests and avoid hospital admission simply because they cannot afford the cost. In addition, in spite of having some of the most advanced hospitals, Lebanon, like many other countries, was not prepared for such pandemic due to the limited number of beds in the ICUs and the significant shortage in ventilators. Therefore, in spite of all efforts to “flatten the curve”, Lebanon has recorded an exponential drastic increase in the daily number of cases and deaths. The reason behind that was the massive explosion that occurred at the Port of Beirut on 4 August 2020. The explosion killed over 200 people, injured more than 6000 others, and left around 300,000 people homeless [14], causing chaos in hospitals as well as a spike in the reported COVID-19-positive cases. Following the explosion, various essential hospitals in the capital were completely destroyed, thus putting more weight on the medical staff, especially the radiology department.

On top of that, due to political problems, Lebanon is suffering from a critically deteriorating economic crisis. The national currency, the Lebanese pound (LBP), is falling stiff against the United States dollar (USD). It lost about 80% of its value, thus causing a severe inflation in the country [15]. With inflation reaching terrifyingly high levels, Lebanon is currently ranked second worldwide in terms of hyperinflation according to the Hanke’s Annual Inflation Rate model [16]. Besides increasing poverty level to 55% (compared to 28% in 2019) [17], this inflation negatively impacts the healthcare system as all products needed are imported in foreign currency (i.e., USD or EUR). Most healthcare institutions in the country are private hospitals and/or medical centers and laboratories. Therefore, there is a huge difficulty in coping with the increasing prices of materials and equipment, thus posing a risk to the staff’s health and safety as institutions administrators can no longer afford adequate and/or good quality PPE and disinfecting/cleaning agents. This increases the threat to healthcare members, especially radiographers, of contracting the virus from the workplace and transmitting it to their family and/or loved ones. The worsening economic situation does not only strike the healthcare systems and staff in Lebanon financially, but also drains them of all energy and hospital beds by escalating the daily number of COVID-19-reported cases. Lebanese people are forced to break all lockdown rules and open their shops/businesses in order to put food on their tables and feed their families, a phenomenon accompanied by the absence of social distancing and precautions, thus reflecting a soar in COVID-19 cases and more pressure on the healthcare system.

Following international guidelines, thoracic imaging, especially chest radiography and chest Computed Tomography (CT), are being laboriously used in all hospitals and imaging centers in Lebanon as powerful tools for the diagnosis, detection of complications, and follow-up of COVID-19 patients [18]. Various studies have proven the importance of chest CT in detecting SARS-CoV-2 in patients with negative reverse transcription polymerase chain reaction (RT-PCR) results [19]. One study involving 1014 patients showed that chest CT scan had a higher sensitivity (97%) compared to RT-PCR [20]. In addition, a recent study proved the practicability of magnetic resonance imaging (MRI) in the detection of pulmonary changes and damage caused by COVID-19, thus proposing a potential radiation-free alternative to chest CT, especially when periodic, repetitive scans are required for follow-up [21]. Furthermore, the newly emerging artificial intelligence (AI)-based algorithms that have the ability to detect COVID-19 pneumonia on chest CT with approximately 90.8% accuracy, 84% sensitivity, and 93% specificity [22], can consequently enhance the paramount role radiographers and medical imaging play during the COVID-19 pandemic.

The huge workload on radiology departments in Lebanon, specifically on Lebanese radiographers or radiologic technologists, due to the ongoing pandemic, as well as the worsening economic situation, have led to many adverse effects such as irregular and disturbed shifts, loss of work, increased radiation exposure, deteriorating mental and physical health. There are currently no studies conducted in the region to point out the critical situation radiographers or radiologic technologists are going through. This study aimed to shed light on what the frontline heroes are going through in the midst of all these unprecedented crises, and evaluate factors associated with stress from contracting the COVID-19 virus from the workplace among radiography technicians.

## 2. Methods

### 2.1. Study Design

A cross-sectional study was conducted among radiographers or radiologic technologists registered in the Lebanese Society of Radiographers (LSR) in multiple hospitals and medical centers all over the country. They were requested to fill out an electronic survey specifically tailored to inquire how the pandemic affected them, their work, and their overall wellbeing, directly and indirectly. The study was conducted from 3 December 2020 until 17 December 2020.

### 2.2. Minimal Sample Size Calculation

On the basis of a population size of 325 active radiography technicians, and a 75.4% expected frequency of workplace-related stress after the outbreak [23], we found that the minimal sample size needed for bivariate and multivariable analysis was 152 according to the Epi-info software (Centers for Disease Control and Prevention, Atlanta, GA, USA) [24].

### 2.3. Questionnaire and Variables

The online survey, proposed in 3 languages (English, Arabic, and French), was distributed to all LSR registered members via the social network platforms (i.e., official WhatsApp groups of the syndicate). The survey was composed of 26 questions organized into 6 sections (general, workplace conditions, health and safety, mental/psychologic, financial, and skill/knowledge development questions). The first section aimed to study the demographical and educational status of the radiographers: age, gender, marital status, degree, and workplace type. The second section aimed to assess the changes made in the workplace: variability in shifts, overall changes in department workload, workflow, protocols, and safety measures (PPE use, disinfection, compulsory mask use). The next section discussed the impact of the virus on technologists’ health and safety with questions evaluating whether they have contracted the virus, its severity, its transmission, and the need for any hospital admission. The following section analyzed the mental or psychologic outcome, with questions inquiring about the presence and severity of workplace-related stress, its impact on them and their family/loved ones, and the support received. The fifth section aimed to determine the financial impact of the pandemic on Lebanese radiographers by including questions concerning the monthly salary, the extent of modifications done to that salary, and whether the radiographers are considering quitting their jobs. The last section estimated a rather positive impact of the pandemic, especially during the lockdown periods and decreased shifts, in terms of skills and/or knowledge development with questions evaluating the use of free time in beneficial, recreational activities and the preferred type of activities.

### 2.4. Statistical Analysis

Descriptive, bivariate, and multivariable statistical analyses were conducted using Statistical Package for the Social Sciences (SPSS) v.25 (Armonk, NY, USA). The quantitative variables were expressed as percentages and comparisons were made using the chi-squared test. A multinomial regression was conducted, taking the stress/worry about contracting COVID-19 from the workplace categories (strongly disagree/disagree, agree/strongly agree, and neutral) as the dependent variable. The neutral group was taken as reference. The Nagelkerke pseudo *R*^2^ values were also calculated to determine the variance explained by each independent variable of the outcome variable. Significance was set at *p* < 0.05.

## 3. Results

A total of 212 survey responses that accounted for 32.5% of overall registered radiologic technologists or 65.3% of active members was received. Out of the three survey languages available, the Arabic language was preferred by almost 46.23% (*n* = 98) of radiographers, then the English language with 39.62% (*n* = 84) submissions, followed by the French language with 14.15% (*n* = 30) submissions (Figure 1). As for the rest of the survey questions, the results of the three survey languages were combined.

### 3.1. General Questions

Responses were received mainly from radiographers that belonged to the 20–29 age group (47.17%) and the 30–39 age group (31.13%), while only 1.89% of radiographers were above 60 years old. Concerning the gender distribution of radiographers, responses were almost equally distributed between males and females, with the percentage of males being slightly greater than that of females (51.42% vs. 48.58%, respectively). As for the marital status, results showed that 52.36% of participants were married, 38.21% were single, 8.02% were engaged, and only 1.41% were divorced. Concerning the highest degree in the field, participants held a T.S./L.T. (technical degrees) in Radiography with 36.32% submissions, 16.98% had a university diploma, 33.96% earned their Bachelor of Science (B.S.) degree, 8.02% had a Master of Science (M.S.) degree, and only 4.72% chose the “other” option and relied mainly on practical experience. Regarding the distribution of work locations, most of the participants (71.23%) worked in private hospitals, 14.62% in imaging centers, and only 10.83% in public hospitals. Table 1 summarizes the questions, answers, and percentages for this section.

### 3.2. Workplace Conditions during the Pandemic

A total of 69.81% of participants agreed that the workload in the department was affected by the pandemic (agree (45.28%), strongly agree (24.53%)). Similarly, the highest percentage of participants (58.49%) agreed (agree (37.74%), strongly agree (20.75%)) that their shift duration and distribution were impacted by the pandemic. While voting for the modality that received the most workload, the most selected choices were CT and X-ray, with 47.87% and 38.53%, respectively. Participants were given the chance to select more than one option resulting in a total of 353 votes, 169 for CT and 136 for X-ray. When asked whether the institution is applying an adapted safety protocol for COVID-19 patients, 67.45% of votes agreed (agree (47.17%), strongly agree (20.28%)). In the same sense, most radiographers agreed that their institution is providing PPE and/or cleaning/disinfecting agents with around 69.81% of the votes (agree (43.87%), strongly agree (25.94%)). Likewise, 88.68% of the participants agreed (agree (32.08%), strongly agree (56.60%)) that their institution is forcing all patients, visitors, and staff to wear face masks. Figure 2 summarizes the questions, answers, and percentages for this section.

### 3.3. Health and Safety

Responses showed that 64.15% of radiographers disagreed (disagree (25.00%), strongly disagree (39.15%)) that their institution is providing regular/periodic, free PCR testing for the staff, while only 25.94% agreed. The highest percentage of participants 74.53% did not contract the virus. Out of the 12.26% radiographers who caught the virus, 61.54% got it from the workplace, 34.62% suffered from mild symptoms, and 92.31% were not admitted to the hospital. Only 30.77% of infected radiologic technologists transmitted the virus to family members/friends/colleagues, while 50.00% did not, and 19.23% were not sure. Table 2 summarizes the questions, answers, and percentages for this section.

### 3.4. Financial Questions

Participants were asked to give an estimate (in LBP) of their original monthly salary provided by the institution, according to the work contract (Figure 3).

While 60.85% of radiologic technologists had no change in their salary, 30.19% disclosed that they had 25–50% or even more than 50% reductions in their salary, whereas 4.24% were not getting paid their monthly salary. Moreover, 61.11% of participants disagreed (disagree (36.42%), strongly disagree (24.69%)) on leaving their job/staying home, while 35.80% had a different opinion. Table 3 summarizes the questions, answers, and percentages for this section.

### 3.5. Mental/Psychological Questions

Concerning mental/psychological questions, 60.85% of radiographers agreed (agree (35.85%), strongly agree (25.00%)) that they were feeling stressed/worried about contracting the virus from the workplace. Similarly, 67.92% agreed (agree (45.75%), strongly agree (22.17%)) that their family members, friends, and/or loved ones were affected by this work-related stress. Furthermore, 43.86% of surveyors disagreed (disagree (20.75%), strongly disagree (23.11%)) that their institution is showing adequate social, psychological, and/or financial care/follow up for staff members who contracted the virus. Half of the participants disagreed (disagree (25.47%), strongly disagree (24.53%)) about thinking/planning to change the field of work and leave the healthcare system; 69.81% of responders agreed (agree (25.00%), strongly agree (44.81%)) about leaving the country to seek a better opportunity abroad, while only 16.04% voted for the opposite. Figure 4 summarizes the questions, answers, and percentages for this section.

### 3.6. Skill/Knowledge Development

More than half of the participants (65.09%) used their free time during the lockdown for skill development and/or knowledge expansion. Numerous options were selected regarding the kind of activities done, and many of the participants chose the “other” option. For simplicity, the kinds of activities done are summarized in the bar graph below (Figure 5).

### 3.7. Bivariate Analysis

A significantly higher percentage of persons who had a neutral opinion about the workload being affected by the pandemic agreed/strongly agreed that they are stressed and worried about contracting COVID-19 from the workplace (Table 4). No significant association was found between all other variables and the stress/worry about contracting COVID-19 from the workplace.

### 3.8. Multivariable Analysis

The results of the regression, taking stress/worry about contracting COVID-19 from the workplace (agree/strongly disagree vs. neutral*) as the dependent variable, showed that having 50 or more years vs. 20–29 years (adjusted odds ratio (aOR) = 9.53; *p* = 0.036) was significantly associated with higher odds of agreeing/strongly agreeing about having stress/worry about contracting COVID-19 from the workplace (Table 5).

None of the variables were significantly associated with disagreeing/strongly disagreeing about having stress/worry about contracting COVID-19 from the workplace compared to neutral.

Variables entered in the model: workload affected by the pandemic, institution forces mask wearing, age categories (Nagelkerke pseudo *R*^2^ = 21.3%); Nagelkerke pseudo *R*^2^ for the variable workload affected by the pandemic = 6.9%; Nagelkerke pseudo *R*^2^ for the variable institution forces mask wearing = 6.4%; Nagelkerke pseudo *R*^2^ for the variable age categories = 6.8%; numbers in bold indicate significant *p*-values (*p* < 0.05).

## 4. Discussion

Radiology, especially chest X-ray and CT examinations, has played a crucial role and proven its effectiveness in the diagnosis and follow-up of pneumonia in COVID-19 patients, in addition to assessing better treatment protocols, including measurement of disease changes and predicting prognosis [25]. Radiologic technologists stand equal to, and side by side with, all doctors and nurses who are fighting hand-in-hand in this pandemic, deserving the title “frontline heroes”. With the radiology department being the primary destination to all Emergency Room (ER) patients suffering from respiratory problems and suspected to be COVID-19-positive, the radiographers’ roles and direct contact with these patients weigh no less than those of their medical colleagues.

This distinctive study is the first of its kind in the region and aimed to assess the direct and indirect impact of the ongoing COVID-19 pandemic on radiographers in a country severely affected by the COVID-19 pandemic, in addition to political and economic crisis. 

Most radiologic technologists in the country are relatively young (belonging to the 20–29 age group), with the genders almost equally distributed between males and females. Institutions all over the country are facing a decrease in imaging volumes, as shown by the large number of radiographers who agreed/strongly agreed that the workload in their department and shifts were impacted. Similarly, many hospitals and imaging centers around the globe have also seen significant drops in non-urgent outpatient visits, imaging, and services, even below baseline values. Nevertheless, thoracic imaging volumes involving X-ray and CT were not severely impacted by the pandemic [26]. The results clearly show the vital role of X-ray and CT scan in this management, as they received the greatest amount of votes regarding the modalities, with most workload reflecting the important role of thoracic imaging in managing patients during the pandemic by detecting signs of COVID-19 pneumonia [27]. Most healthcare systems are implementing and adapted safety protocol when dealing with COVID-19 patients and are supplying the staff with the appropriate PPEs (i.e., masks, face shields, gloves, etc.) and cleaning/disinfecting agents. The strict rules concerning the compulsory use of face masks by patients, visitors, and staff also reflect the efficiency of institutions’ efforts in controlling disease spread, a vital strategy that is implemented around the globe for disease handling and protection of the staff from infections [28]. Proven effective, a very high number of participants did not contract the virus. Those who did, on the other hand, suffered from mild to moderate symptoms and did not require hospital admission.

As for the financial aspect, the median monthly income of the Lebanese radiographers is 1661,125 LBP, which is relatively low compared to the high hyperinflation levels the country is suffering from and the fall of the Lebanese currency compared to foreign currencies, especially in black markets. The radiographer’s average monthly salary that used to be equivalent to around 1096 USD pre-hyperinflation (official exchange rate 1 USD = 1515 LBP) now barely equals 144 USD (black market 1 USD = 11,500 LBP). Despite the economic crisis, more than half of the participants reported no change in their monthly income, yet a considerable number of radiographers are suffering from a 20 to 50% reduction in their salary. A struggle that reflects the fact that the middle-income level of the Lebanese population is shrinking. Regardless of all challenges and reductions, radiographers disagreed quitting their job and/or staying home, as they strongly hold onto their humane role and life-saving duties at all costs. The proceeding pandemic has affected Lebanese radiographers, not only physically, but mentally as well, with most radiologic technologists reporting increased stress and/or anxiety regrading contracting the virus from the workplace. This stress has also impacted their families, friends, and/or loved ones too. This stress and “burnout” does not only apply to Lebanese radiographers and their families, as it is also reported by countless radiographers universally who most suffer constant distress concerning the high risk of contracting the virus from COVID-19-positive patients when exposed to them without appropriate PPE [29]. Various studies and publications have heavily discussed how stress, especially at times of a pandemic such as COVID-19, may be a valuable variable in preventing the spread of the virus. Those suffering constant stress, mainly the elderly as shown in the statistical analysis, have an aggravated fear of getting infected by the virus, added to the fear of believing false information found online. These fears have a positive impact on these individuals, forcing them to constantly search for legitimate and scientific information and avoid false theories. Moreover, this “beneficial stress” may lead to the correct and efficient application of protection methods and the adoption of correct hygienic practices. However, the adverse effects of stress, especially over prolonged period the pandemic has been going as well as the forced lockdowns and frightening death toll, may outweigh the benefits. Prolonged exposure to stress can negatively impact body function, leading to morphological and functional changes in various parts of the brain like such as hippocampus (leading to memory and learning disorders), as well as amygdala and temporal lobe (resulting in cognitive, behavioral, and mood disorders). In addition, it was proven that stress can impair the immune system, thus leading to more illnesses [30]. Lebanese radiographers are passing through a highly stressful period, whether it be through physical, mental, financial, and/or work-related strain. Unfortunately, institutions in Lebanon do not show adequate social, psychological, and/or financial care or follow-up to the infected staff members, which, in turn, add up to the deteriorating mental health of radiographers in the country. In spite of that, they disagree on changing their field of work and leaving the healthcare system but, in turn, are planning to leave the country in hopes of seeking a better opportunity abroad.

In defiance of all the negative direct and indirect impacts COVID-19 had on the Lebanese radiographers, a significant number benefited from their free time during the lockdown to develop their skills, expand their knowledge, and engage in positive recreational activities. With learning a new language, developing a hobby, getting new certificates, and reading being the most frequent types of activities reported by the participants.

## 5. Conclusions

The unprecedented COVID-19 pandemic has adversely impacted the world—travel, tourism, the economy, etc., were all shaken and frozen. The greatest impact, however, struck the unprepared healthcare systems that are struggling with the rapid spread of this virus. Worn out and exhausted, frontline heroes do not hesitate when it comes to emergency situations and always answer their humane call. With the heavy workload falling on members of the radiology department, radiographers and radiologists had to cope with irregular shifts, increased radiation exposure, and serious risk of contracting the virus. Following international guidelines and adapting some of their own, the Lebanese radiographers spare no efforts when it comes to narrowing the viral spread in the country and saving lives. However, it is clear that radiographers in Lebanon are suffering from deteriorating physical and psychological health due to the constant stress, absence of any support from their healthcare instructions, added to the country’s collapsing economy and increasing hyperinflation levels. With such tension and strain, not only the radiographers but any other member in the healthcare system passing through the same conditions cannot continue with the same productivity, motivation, enthusiasm, and power, and thus will collapse later on. This cross-sectional study highlighted all the obstacles, dangers, and challenges that radiographers or radiologic technologists in Lebanon are combatting whether physically, mentally, and/or financially. It also aims to provide better and safer workplace conditions together with more specifically adapted and robust workflow to fight not only the ongoing pandemic, but also future ones as well. Finally, with radiology being the “eye of medicine”, radiographers should be appreciated just like other healthcare members as the pandemic warriors that wear lead aprons instead of capes.

## Figures and Tables

**Figure 1 healthcare-09-00362-f001:**
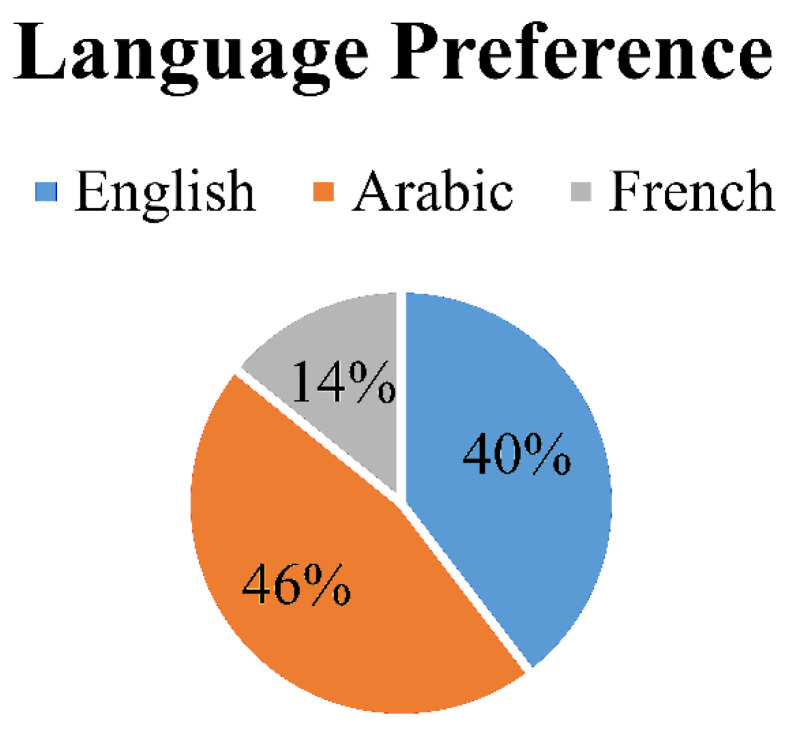
Pie chart showing the percentage distribution of the three languages: English, Arabic, and French (the percentages were rounded to the nearest whole number).

**Figure 2 healthcare-09-00362-f002:**
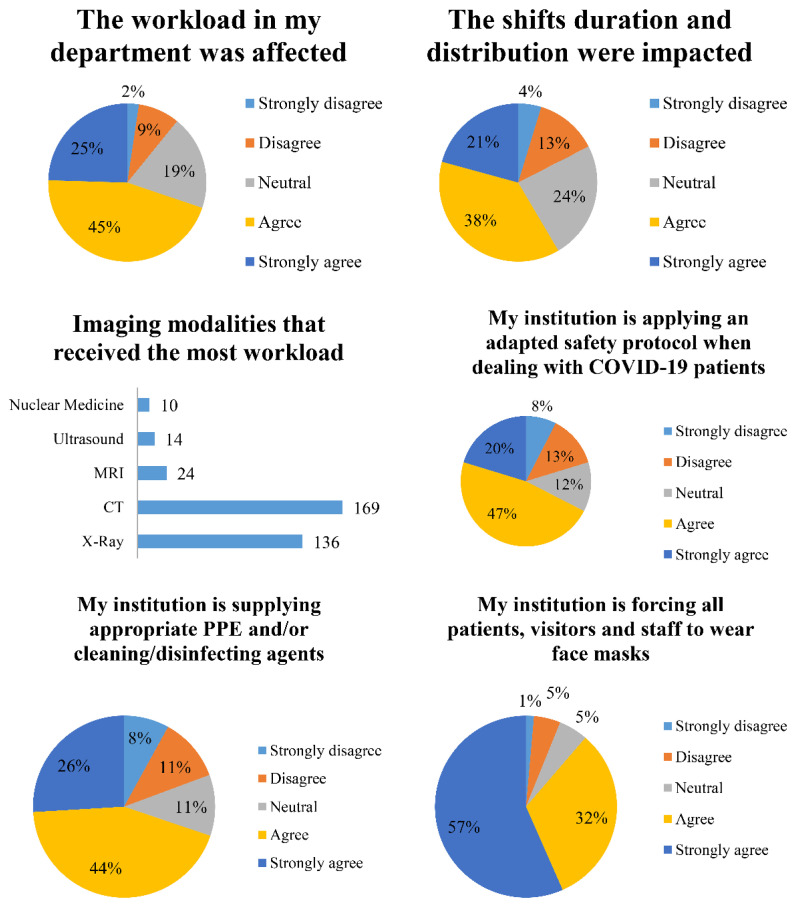
Pie charts summarizing the questions, choices, and percentages concerning the workplace conditions section.

**Figure 3 healthcare-09-00362-f003:**
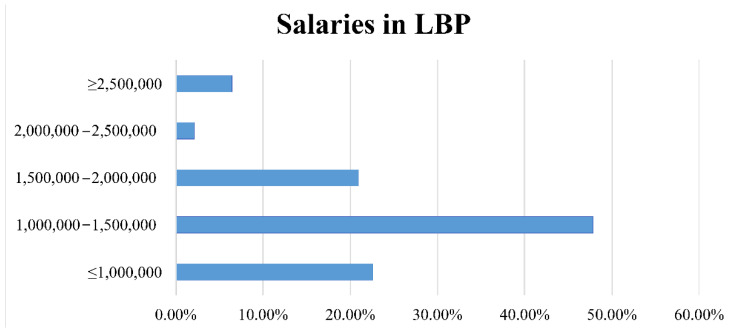
Bar graph presenting the percentages of salary ranges as disclosed by the participants.

**Figure 4 healthcare-09-00362-f004:**
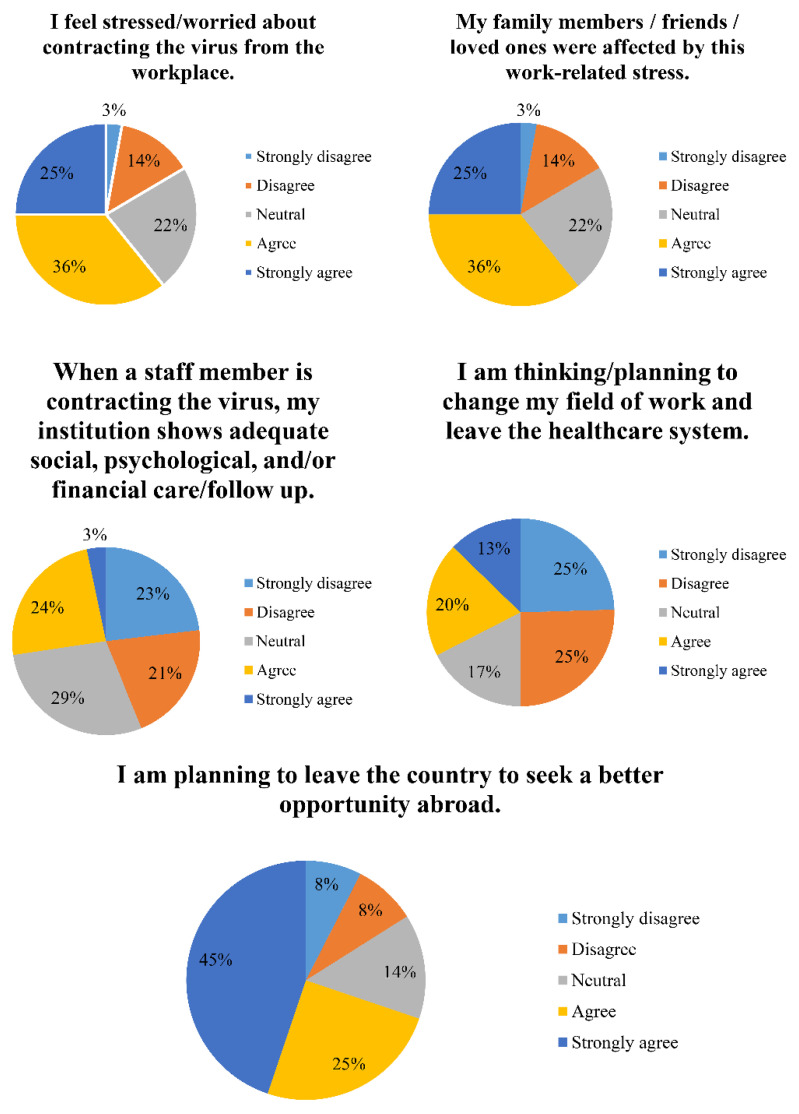
Pie charts summarizing the questions, choices, and percentages concerning the mental/psychological questions section.

**Figure 5 healthcare-09-00362-f005:**
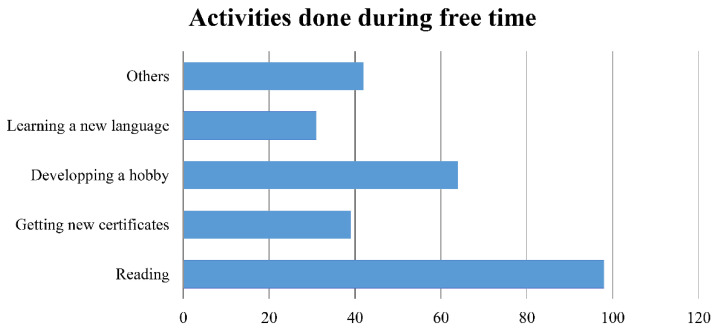
Bar graph summarizing the various types of recreational activities done by the participants during their free time in the lockdown.

**Table 1 healthcare-09-00362-t001:** Table summarizing the questions, choices, and percentages concerning the general questions section.

Question			Choices and Percentages		
Age	20–29 47.17%	30–39 31.13%	40–49 11.79%	50–59 8.02%	60+ 1.89%
Gender	Males 51.42%			Females 48.58%	
Marital Status	Single 38.21%		Engaged 8.02%	Married 52.36%	Divorced 1.41%
Highest Degree	T.S./L.T. 36.32%	Diploma 16.98%	B.S. 33.96%	M.S. 8.02%	Others 4.72%
Work Location	Private Hospital 71.23%		Public Hospital 10.85%	Lab/Medical Imaging Center 14.62%	Others 3.30%

**Table 2 healthcare-09-00362-t002:** Table summarizing the questions, choices, and percentages concerning the health and safety section.

Question	Choices and Percentages				
My institution is providing regular/periodic, free PCR testing for staff.	Strongly Disagree 39.15%	Disagree 25.00%	Neutral 9.91%	Agree 16.04%	Strongly Agree 9.90%
Have you contracted the virus?	Yes 12.26%		No 74.53%		I am not sure 13.21%
Is it from the workplace?	Yes 61.54%		No 11.54%		I am not sure 26.92%
What was the severity of the disease?	No symptoms 7.69%	Mild symptoms 34.62%		Moderate symptoms 30.77%	Severe symptoms 26.92%
Were you admitted to the hospital?	Yes 8.02%			No 25.94%	
Did you transmit the virus to any family member/friend/colleague?	Yes 30.77%		No 50.00%		I am not sure 19.23%

**Table 3 healthcare-09-00362-t003:** Table summarizing the questions, choices, and percentages concerning the financial questions section.

Question	Choices and Percentages				
To what extent did the institution modify/decrease the monthly salary provided to you in accordance with the economic situation?	Severely (>50% reduction) 6.13%	Moderately (25–50% reduction) 24.06%	No Change 60.85%	My salary was modified but what was reduced will be paid later on 4.72%	I am not getting paid my monthly salary 4.24%
I am considering leaving job/staying home, as it is not worth it.	Strongly Disagree 24.69%	Disagree 36.42%	Neutral 3.09%	Agree 25.31%	Strongly Agree 10.49%

**Table 4 healthcare-09-00362-t004:** Bivariate analysis of factors associated with stress categories.

Variable	Stress/Worry about Contracting COVID-19 from the Workplace			*p*
	Neutral	Strongly Disagree/Disagree	Agree/Strongly Agree	
**Age categories (in years)**				0.145
20–29	27 (26.5%)	22 (21.6%)	53 (52.0%)	
30–39	16 (24.2%)	9 (13.6%)	41 (62.1%)	
40–49	4 (16.0%)	3 (12.0%)	18 (72.0%)	
50 and above	1 (4.8%)	3 (14.3%)	17 (81.0%)	
Gender				0.238
Male	29 (26.6%)	20 (18.3%)	60 (55.0%)	
Female	19 (18.1%)	17 (16.2%)	69 (65.7%)	
**Marital status**				0.841
Single/engaged/divorced	25 (24.3%)	17 (16.5%)	61 (59.2%)	
Married	23 (20.7%)	20 (18.0%)	68 (61.3%)	
**Workload affected by the pandemic**				**0.034**
Strongly disagree/disagree	12 (29.3%)	6 (14.6%)	23 (56.1%)	
Neutral	0 (0%)	5 (21.7%)	18 (78.3%)	
Agree/strongly agree	36 (24.0%)	26 (17.3%)	88 (58.7%)	
**Shift duration distribution impacted during the pandemic**				0.204
Strongly disagree/disagree	12 (23.5%)	4 (7.8%)	35 (68.6%)	
Neutral	8 (21.6%)	5 (13.5%)	24 (64.9%)	
Agree/strongly agree	28 (22.2%)	28 (22.2%)	70 (55.6%)	
**Institution applies adapted safety protocol**				0.268
Strongly disagree/disagree	4 (15.4%)	2 (7.7%)	20 (76.9%)	
Neutral	8 (18.6%)	6 (14.0%)	29 (67.4%)	
Agree/strongly agree	36 (24.8%)	29 (20.0%)	80 (55.2%)	
**Institution supplying cleaning agents**				0.224
Strongly disagree/disagree	4 (17.4%)	4 (17.4%)	15 (65.2%)	
Neutral	6 (14.6%)	4 (9.8%)	31 (75.6%)	
Agree/strongly agree	38 (25.3%)	29 (19.3%)	83 (55.3%)	
**Institution forces mask wearing**				0.083
Strongly disagree/disagree	4 (36.4%)	0 (0%)	7 (63.6%)	
Neutral	0 (0%)	2 (15.4%)	11 (84.6%)	
Agree/strongly agree	44 (23.2%)	35 (18.4%)	111 (58.4%)	

Numbers in bold indicate significant *p*-values.

**Table 5 healthcare-09-00362-t005:** Multinomial regression.

Stress/Worry About Contracting COVID-19 from the Workplace (Agree/Strongly Agree vs. Neutral)			
Variable	aOR	*p*	95% CI
Age categories (in years)		0.182	
20–29	1		
30–39	1.28	0.531	0.59–2.79
40–49	1.69	0.407	0.49–5.81
50 and above	9.53	**0.036**	1.16–78.30

Numbers in bold indicate significant *p*-values.

## Data Availability

The data presented in this study are available on request from the corresponding author.

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
