# Peer review of "Shedding Light on the Direct and Indirect Impact of the COVID-19 Pandemic on the Lebanese Radiographers or Radiologic Technologists: A Crisis within Crises"

_healthcare, 2021, doi:10.3390/healthcare9030362_

Round 1

Reviewer 1 Report

This article addresses the important issue of factors affecting well-being of radiographers in Lebanon.

Unfortunately, the study in its current form is just descriptive.  It is imperative that the authors perform statistical comparisons between variables of a given section and also formulate statistical models to extract information on variables that have a significant effect on some of the factors considered such as mental health, working conditions etc;

With regards to the important factor of stress, please discuss about recent studies emphasizing the role of this variable in models of covid spreading and long term effect on mental health, e.g.    Stress as a meaningful variable in models of covid-19 spreading. https://doi.org/10.31234/osf.io/kcpqm

Reviewer 2 Report

 Introduction

Page 2, lines 61-62. “[Lebanon] …. has the best healthcare level in the region. “ According to WHO, Lebanon ranks 91st out of 191 countries1 in overall health system performance. I suggest the authors include a reference for their claim.

Methods

Page 3, line 120. A review of the methods points that the design of this study is simply a cross-sectional study, not a “prospective cross-sectional study” because the study was conducted at a point in time and does not involve follow-up over a course of time.  I suggest that this be corrected.

What is/are the research question(s)?

What are the objectives of the study?

What is the validity and reliability of each of the survey instruments?

What is the study population? Sampling method? and how was the sample size determined?

What is the outcome variable? what are the independent variables?

Result

Apart from a descriptive analysis of sociodemographic variables, it is not clear what research questions were addressed by this study.  There is no clear indication of what is the outcome variable and what are the independent variables.  There were no inferential statistics involved. There was no pre-pandemic baseline data to compare, therefore, the study does not provide conclusive evidence about how much of the variation in each variable, when applicable, could be attributed to the Covid-19 epidemic and how much of the variation in each variable could have happened in the absence of COVID-19.

The study does not quantitatively or qualitatively (through interviews) demonstrate that radiologists were more affected than other frontline workers such as nurses, doctors, and other front-line care providers. Simply stating radiologists had a lot of work to do does not imply they had more work to do than other frontline workers unless we demonstrate it through a statistical significance accounting for as many variables as possible.

Conclusion

The conclusion does not produce new knowledge or identify any gap in knowledge or prove that care-providers in one department are statistically significantly more affected than other departments in the same health care facility or other health-care facilities.

Reference

Tandon, A., Murray, C. J., Lauer, J. A., & Evans, D. B. (2000). Measuring overall health system performance for 191 countries. Geneva: World Health Organization.

Round 2

Reviewer 1 Report

-Please proof read the manuscript, there are still numerous typos and grammatical errors.

-Please follow proper format of citations and references as specified by the journal.

-Please provide the web site of the Epi info software.

-The authors mentioned that the online-survey was provided via social network platforms, please specify what social networks were used. Was data privacy assured? How was the data retrieved?

-Please specify the software that was utilized for statistics.

Reviewer 2 Report

The authors have adequately responded to most of this reviewer’s concerns and should be commended for including some inferential data analysis.

Still outstanding issues:

  1. It is not possible to compare a Bloomberg ranking with a WHO ranking to claim the improvement of standing for Lebanon’s health care system. As a matter of fact, the same ranking made at different times should be used to assess any trends. I suggest that the authors underscore the decline in ranking for Lebanon as stated in the 2020 Bloomberg’s Healthcare Efficiency Index compared to an earlier ranking by the same firm( 48th in 2020 versus 23rd in 2018) so that this is brought to the attention of the country’s health authorities (see the link: https://www.bloomberg.com/news/articles/2020-12-18/asia-trounces-u-s-in-health-efficiency-index-amid-pandemic)
  2. The response given by the authors about determining the validity and reliability of survey instruments is simply erroneous. One way of finding out the validity and reliability of survey instruments to determine their Cronbach’s alpha. Please do it. 
  3. The authors should also determine the variance explained by each statistically significant independent variable of the outcome variable.

Author Response

This manuscript is a resubmission of an earlier submission. The following is a list of the peer review reports and author responses from that submission.